# Model Similarity Mitigates Test Set Overuse

**Horia Mania**
UC Berkeley
hmania@berkeley.edu

**John Miller**
UC Berkeley
miller_john@berkeley.edu

**Ludwig Schmidt**
UC Berkeley
ludwig@berkeley.edu

**Moritz Hardt**
UC Berkeley
hardt@berkeley.edu

**Benjamin Recht**
UC Berkeley
brecht@berkeley.edu

## Abstract

Excessive reuse of test data has become commonplace in today's machine learning workflows. Popular benchmarks, competitions, industrial scale tuning, among other applications, all involve test data reuse beyond guidance by statistical confidence bounds. Nonetheless, recent replication studies give evidence that popular benchmarks continue to support progress despite years of extensive reuse. We proffer a new explanation for the apparent longevity of test data: Many proposed models are similar in their predictions and we prove that this similarity mitigates overfitting. Specifically, we show empirically that models proposed for the ImageNet ILSVRC benchmark agree in their predictions well beyond what we can conclude from their accuracy levels alone. Likewise, models created by large scale hyperparameter search enjoy high levels of similarity. Motivated by these empirical observations, we give a non-asymptotic generalization bound that takes similarity into account, leading to meaningful confidence bounds in practical settings.

## 1 Introduction

Be it validation sets for model tuning, popular benchmark data, or machine learning competitions, the holdout method is central to the scientific and industrial activities of the machine learning community. As compute resources scale, a growing number of practitioners evaluate an unprecedented number of models against various holdout sets. These practices, collectively, put significant pressure on the statistical guarantees of the holdout method. Theory suggests that for $k$ models chosen independently of $n$ test data points, the holdout method provides valid risk estimates for each of these models up to a deviation on the order of $\sqrt{\log(k)/n}$ [5]. But this bound is the consequence of an unrealistic assumption. In practice, models incorporate prior information about the available test data since human analysts choose models in a manner guided by previous results. Adaptive hyperparameter search algorithms similarly evolve models on the basis of past trials [12].

Adaptivity significantly complicates the theoretical guarantees of the holdout method. A simple adaptive strategy, resembling the practice of selectively ensembling $k$ models, can bias the holdout method by as much as $\sqrt{k/n}$ [5]. If this bound were attained in practice, holdout data across the board would rapidly lose its value over time. Nonetheless, recent replication studies give evidence that popular benchmarks continue to support progress despite years of extensive reuse [15, 20].

In this work, we contribute a new explanation for why the adaptive bound is not attained in practice and why even the standard non-adaptive bound is more pessimistic than it needs to be. Our explanation centers around the phenomenon of *model similarity*. Practitioners evaluate models that incorporate common priors, past experiences, and standard practices. As we show empirically, this

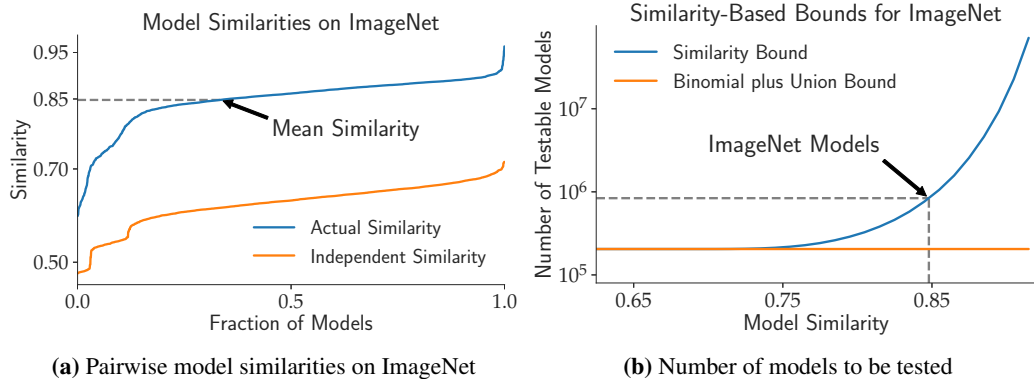

**(a)** Pairwise model similarities on ImageNet

**(b)** Number of models to be tested

Figure 1: **(a)** shows the empirical pairwise similarity between Imagenet models and the hypothetical similarity between models if they were making mistakes independently. **(b)** plots the number of testable models on Imagenet such that the population error rates for all models are estimated up to $\pm 1\%$ error with probability 0.95. We compare the guarantee of the standard union bound with that of a union bound which considers model similarities.

results in models that exhibit significant agreement in their predictions, well beyond what would follow from their accuracy values alone. Complementing our empirical investigation of model similarity, we provide a new theoretical analysis of the holdout method that takes model similarity into account, vastly improving over known bounds in the adaptive and non-adaptive cases when model similarity is high.

## 1.1 Our contributions

Our contributions are two-fold. On the empirical side, we demonstrate that a large number of proposed ImageNet [3, 16] and CIFAR-10 [9] models exhibit a high degree of similarity: Their predictions agree far more than we would be able to deduce from their accuracy levels alone. Complementing our empirical findings, we give new generalization bounds that incorporate a measure of similarity. Our generalization bounds help to explain why holdout data has much greater longevity than prior bounds suggest when models are highly similar, as is the case in practice. Figure 1 summarizes these two complementary developments.

Underlying Figure 1a is a family of representative ImageNet models whose pairwise similarity we evaluate. The mean level of similarity of these models, together with a refined union bound, offers a $4\times$ improvement over a carefully optimized baseline bound that does not take model similarity into account. In Figure 1b we compare our guarantee on the number of holdout reuses with the baseline bound. This illustrates that our bound is not just asymptotic, but concrete—it gives meaningful values in the practical regime. Moreover, in Section 5 we discuss how an additional assumption on model predictions can boost the similarity based guarantee by multiple orders of magnitude.

Investigating model similarity in practice further, we evaluate similarity of models encountered during the course of a large random hyperparamter search and a large neural architecture search for the CIFAR-10 dataset. We find that the pairwise model similarities throughout both procedures remain high. The similarity provides a counterweight to the massive number of model evaluations, limiting the amount of overfitting we observe.

## 1.2 Related work

Recht et al. [15] recently created new test sets for ImageNet and CIFAR10, carefully following the original test set creation processes. Reevaluating all proposed models on the new test sets showed that while there was generally an absolute performance drop, the effect of overfitting due to adaptive behavior was limited to non-existent. Indeed, newer and better models on the old test set also performed better on the new test set, even though they had in principle more time to adapt to the test set. Also, Yadav and Bottou [20] recently released a new test set for the seminal MNIST task, on which they observed no overfitting.

Dwork et al. [5] recognized the issue of adaptivity in holdout reuse and provided new holdout mechanisms based on noise addition that support quadratically more queries than the standard method in the worse case. There is a rich line of work on adaptive data analysis; Smith [18] offers a comprehensive survey of the field.

We are not the first to proffer an explanation for the apparent lack of overfitting in machine learning benchmarks. Blum and Hardt [2] argued that if analysts only check if they improved on the previous best model, while ignoring models that did not improve, better adaptive generalization bounds are possible. Zrnic and Hardt [21] offered improved guarantees for adaptive analysts that satisfy natural assumptions, e.g. the analyst is unable to arbitrarily use information from queries asked far in the past. More recently, Feldman et al. [6] gave evidence that the number of classes in a classification problem helps mitigate overfitting in benchmarks. We see these different explanations as playing together in what is likely the full explanation of the available empirical evidence. In parallel to our work, Yadav and Bottou [20] discussed the advantages of comparing models on the same test set; pairing tests can provide tighter confidence bounds for model comparisons in this setting than individual confidence intervals for each model.

## 2 Problem setup

Let $f : \mathcal{X} \to \mathcal{Y}$ be a classifier mapping examples from domain $\mathcal{X}$ to a label from the set $\mathcal{Y}$. Moreover, we consider a test set $S = \{(x_1, y_1), \ldots\}$ of $n$ examples sampled i.i.d. from a data distribution $\mathcal{D}$. The main quantity we aim to analyze is the gap between the accuracy of the classifier $f$ on the test set $S$ and the population accuracy of the same classifier under the distribution $\mathcal{D}$. If the gap between the two accuracies is large, we say $f$ overfit to the test set.

As is commonly done in the adaptive data analysis literature [1], we formalize interactions with the test set via *statistical queries* $q : \mathcal{X} \times \mathcal{Y} \to \mathbb{R}$. In our case, the queries are $\{0, 1\}$-valued; given a classifier $f$ we consider the query $q_f$ defined by $q_f(z) = \mathbb{1}\{f(x) \neq y\}$, where $z = (x, y)$. Then, we denote the empirical mean of query $q_f$ on the test set $S$ (i.e., $f$'s test error) by $\mathbb{E}_S[q_f] = \frac{1}{n}\sum_{i=1}^{n} q_f(z_i)$. The population mean (population error) is accordingly defined as $\mathbb{E}_\mathcal{D}[q] = \mathbb{E}_{z \sim \mathcal{D}} q(z)$.

When discussing overfitting, we are usually interested in a set of classifiers, e.g., obtained via a hyperparameter search. Let $f_1, \ldots, f_k$ be such a set of classifiers and $q_1, \ldots, q_k$ be the set of corresponding queries. To quantify the probability that overfitting occurs (i.e., one of the $f_i$ has a large deviation between test and population accuracy), we would like to upper bound the probability

$$\mathbb{P}\left(\max_{1 \leq i \leq k} |\mathbb{E}_S[q_i] - \mathbb{E}_\mathcal{D}[q_i]| \geq \varepsilon\right). \tag{1}$$

A standard way to bound (1) is to invoke the union bound and treat each query separately:

$$\mathbb{P}\left(\max_{1 \leq i \leq k} |\mathbb{E}_S[q_i] - \mathbb{E}_\mathcal{D}[q_i]| \geq \varepsilon\right) \leq \sum_{i=1}^{k} \mathbb{P}\left(|\mathbb{E}_S[q_i] - \mathbb{E}_\mathcal{D}[q_i]| \geq \varepsilon\right) \tag{2}$$

We can then utilize standard concentration results to bound the right hand side. However, such an approach inherently cannot capture dependencies between the queries $q_i$ (or classifiers $f_i$). In particular, we are interested in the similarity between two queries $q$ and $q'$ measured by $\mathbb{P}(q(z) = q'(z))$ (the probability of agreement between the 0-1 losses of the corresponding two classifiers). The main goal of this paper is to understand how high similarity can lead to better bounds on (1), both in theory and in numerical experiments with real data from ImageNet and CIFAR-10.

## 3 Non-adaptive classification

We begin by analyzing the effect of the classifier similarity when the classifiers to be evaluated are chosen *non-adaptively*. For instance, this is the case when the algorithm designer fixes a grid of hyperparameters to be explored before evaluating any of the classifiers on the test set. To draw valid gains from the hyperparameter search, it is important that the resulting test accuracies reflect the true population accuracies, i.e., probability (1) is small.

Bound (2) is sharp when the events $\{|\mathbb{E}_S[q_i] - \mathbb{E}_\mathcal{D}[q_i]| \geq \varepsilon\}$ are almost disjoint, which is not true when the queries are similar to each other. To address this issue, we modify our use of the union

bound. We consider the left tails $\mathcal{E}_i = \{\mathbb{E}_S[q_i] - \mathbb{E}_\mathcal{D}[q_i] \geq \varepsilon\}$. For any $t \geq 0$, we obtain

$$\mathbb{P}\left(\bigcup_{i=1}^{k} \mathcal{E}_i\right) \leq \mathbb{P}\left(\{\mathbb{E}_S[q_1] - \mathbb{E}_\mathcal{D}[q_1] \geq \varepsilon - t\} \bigcup_{i=2}^{k} \mathcal{E}_i\right) \tag{3}$$

$$= \mathbb{P}\left(\mathbb{E}_S[q_1] - \mathbb{E}_\mathcal{D}[q_1] \geq \varepsilon - t\right) + \mathbb{P}\left(\bigcup_{i=2}^{k} \mathcal{E}_i \cap \{\mathbb{E}_S[q_1] - \mathbb{E}_\mathcal{D}[q_1] < \varepsilon - t\}\right)$$

$$\leq \mathbb{P}\left(\mathbb{E}_S[q_1] - \mathbb{E}_\mathcal{D}[q_1] \geq \varepsilon - t\right) + \sum_{i=2}^{k} \mathbb{P}\left(\mathcal{E}_i \cap \{\mathbb{E}_S[q_1] - \mathbb{E}_\mathcal{D}[q_1] < \varepsilon - t\}\right).$$

Intuitively, the terms $\mathbb{P}\left(\mathcal{E}_i \cap \{\mathbb{E}_S[q_1] - \mathbb{E}_\mathcal{D}[q_1] < \varepsilon - t\}\right)$ are small when the queries $q_1$ and $q_i$ are similar: if $\mathbb{P}(q_1(z) = q_i(z))$ is large, we cannot simultaneously have $\mathbb{E}_S[q_1] < \mathbb{E}_\mathcal{D}[q_1] + \varepsilon - t$ and $\mathbb{E}_S[q_i] \geq \mathbb{E}_\mathcal{D}[q_i] + \varepsilon$ since the deviations go into opposite directions. In the rest of this section, we make this intuition precise in and derive an upper bound on (1) in terms of the query similarities. Before we state our main result, we introduce the following notion of a similarity covering.

**Definition 1.** *Let $\mathcal{F}$ be a set of queries. We say a query set $M$ is a $\eta$ similarity cover of $\mathcal{F}$ if for any query $q \in \mathcal{F}$ there exist $q', q'' \in M$ such that $\mathbb{E}_\mathcal{D}[q'] \leq \mathbb{E}_\mathcal{D}[q]$, $\mathbb{E}_\mathcal{D}[q''] \geq \mathbb{E}_\mathcal{D}[q]$, $\mathbb{P}(q'(z) = q(z)) \geq \eta$, and $\mathbb{P}(q''(z) = q(z)) \geq \eta$ ( $M$ does not necessarily have to be a subset of $\mathcal{F}$). Let $N_\eta(\mathcal{F})$ denote the size of a minimal $\eta$ similarity cover of $\mathcal{F}$ (when the query set $\mathcal{F}$ is clear from context we use the simpler notation $N_\eta$).*

**Theorem 2.** *Let $\mathcal{F} = \{q_1, q_2, \ldots, q_k\}$ be a collection of queries $q_i \colon \mathcal{Z} \to \{0, 1\}$ independent of the test set $\{z_1, z_2, \ldots, z_n\}$. Then, for any $\eta \in [0, 1]$ we have*

$$\mathbb{P}\left(\max_{1 \leq i \leq k} |\mathbb{E}_S[q_i] - \mathbb{E}_\mathcal{D}[q_i]| \geq \varepsilon\right) \leq 2N_\eta e^{-\frac{n\varepsilon^2}{2}} + 2k e^{-\frac{n\varepsilon}{4} \log\left(1 + \frac{\varepsilon}{4(1-\eta)}\right)}. \tag{4}$$

*Then, for all $\eta \leq 1 - \max\left\{\frac{2\log(4k/\delta)}{n}, \sqrt{\frac{\log(4N_\eta/\delta)}{2n}}\right\}$, we have with probability $1 - \delta$*

$$\max_{1 \leq i \leq k} |\mathbb{E}_S[q_i] - \mathbb{E}_\mathcal{D}[q_i]| \leq \max\left\{\sqrt{\frac{2\log(4N_\eta/\delta)}{n}}, \sqrt{\frac{32(1-\eta)\log(4k/\delta)}{n}}\right\}. \tag{5}$$

*Moreover, if $\varepsilon = \sqrt{\frac{\log((2N_\eta+1)/\delta)}{n}}$ and $\eta \geq 1 - \frac{\varepsilon}{4\left(e^{2\varepsilon}(2k)^{\frac{4}{n\varepsilon}} - 1\right)}$, we have with probability $1 - \delta$*

$$\max_{1 \leq i \leq k} |\mathbb{E}_S[q_i] - \mathbb{E}_\mathcal{D}[q_i]| \leq \varepsilon. \tag{6}$$

To elucidate how model similarity $\eta$ controls the number of queries $k$ for which Theorem (2) gives a non-trivial bound, consider the case where $N_\eta = 1$, i.e. at least one model is $\eta$-similar to all of the others. As the similarity $\eta$ of the model collection grows, the number of queries $k$ grows as well, as the following simple result shows.

**Corollary 3.** *Let $\mathcal{F} = \{q_1, q_2, \ldots, q_k\}$ be a collection of $k$ queries $q_i \colon \mathcal{Z} \to \{0, 1\}$ fixed independently of the test set. Choose $\eta_\star$ so that $N_{\eta_\star} = 1$. Suppose $n \geq c_1 \max\{\varepsilon^{-1}, \varepsilon^{-2}\}$ and the number of queries $k$ satisfies*

$$k \leq \frac{c_2 \varepsilon}{(1 - \eta_\star)} \tag{7}$$

*for positive constants $c_1, c_2$. Then, with probability $3/4$, $\max_{1 \leq i \leq k} |\mathbb{E}_S[q_i] - \mathbb{E}_\mathcal{D}[q_i]| \leq \varepsilon$.*

The proof of Theorem (2) starts with the refined union bound (3), or a standard triangle inequality, and then applies the Chernoff concentration bound shown in Lemma 4 for random variables which take values in $\{-1, 0, 1\}$. We defer the proof details of both the lemma and the theorem to Appendix A.

**Lemma 4.** *Suppose $X_i$ are i.i.d. discrete random variables which take values $-1$, $0$, and $1$ with probabilities $p_{-1}$, $p_0$, and $p_1$ respectively, and hence $\mathbb{E}X_i = p_1 - p_{-1}$. Then, for any $t \geq 0$ such that $p_1 - p_{-1} + t/2 \geq 0$ we have*

$$\mathbb{P}\left(\frac{1}{n}\sum_{i=1}^{n} X_i > p_1 - p_{-1} + t\right) \leq e^{-\frac{nt}{2} \log\left(1 + \frac{t}{2p_1}\right)}.$$

Discretization arguments based on coverings are standard in statistical learning theory. Covers based on the population Hamming distance $\mathbb{P}(q'(z) \neq q(z))$ have been previously studied [4, 11] (Note that for $\{0, 1\}$-valued queries the Hamming distance is equal to the $L^2$ and $L^1$ distances). An important distinction between our result and prior work is that prior work requires $\eta$ to be greater than $1 - \varepsilon$. Theorem 2 can offer an improvement over the standard guarantee $\sqrt{\log(k)/n}$ even when $\eta$ is much smaller than $1 - \varepsilon$. First of all note that (5) holds for $\eta$ bounded away from one. Moreover, since $e^{2\epsilon} \approx 1 + 2\epsilon$, if $(2k)^{\frac{4}{n\varepsilon}} \leq 1 + \sqrt{\varepsilon}$ (the choice of $1 + \sqrt{\varepsilon}$ is somewhat arbitrary), we see the requirement on $\eta$ for (6) is satisfied when $\eta$ is on the order of $1 - \sqrt{\varepsilon}$.

## 4 Adaptive classification

In the previous section, we showed similarity can prevent overfitting when the sequence of queries is chosen *non-adaptively*, i.e. when the queries $\{q_1, q_2, \ldots, q_n\}$ are fixed independently of the test set $S$. In the *adaptive* setting, we assume the query $q_t$ can be selected as a function of the previous queries $\{q_1, q_2, \ldots, q_{t-1}\}$ and estimates $\{\mathbb{E}_S[q_1], \mathbb{E}_S[q_2], \ldots, \mathbb{E}_S[q_{t-1}]\}$. Even when queries are chosen adaptively, we show leveraging similarity can provide sharper bounds on the probability of overfitting, $\mathbb{P}\left(\max_{1 \leq i \leq k} |\mathbb{E}_S[q_i] - \mathbb{E}_{\mathcal{D}}[q_i]| \geq \varepsilon\right)$.

In the adaptive setting, the field of adaptive data analysis offers a rich technical repertoire to address overfitting [5, 18]. In this framework, analogous to the typical machine learning workflow, an analyst iteratively selects a classifier and then queries a mechanism to provide an estimate of test-set performance. In practice, the mechanism often used is the *Trivial Mechanism* which computes the empirical mean of the query on the test set and returns the exact value to the analyst. For simplicity, we study how similarity improves the performance of the trivial mechanism.

The empirical mean of any query can take at most $n + 1$ values, and thus a deterministic analyst might ask at most $(n + 1)^{k-1}$ queries in $k$ rounds of interaction with the Trivial Mechanism. Let $\mathcal{F}$ denote the set of $(n + 1)^{k-1}$ possible queries. Then, we apply Theorem 2 to $\mathcal{F}$.

**Corollary 5.** *Let $\mathcal{F}$ be the set of queries that a fixed analyst $\mathcal{A}$ might query the Trivial Mechanism. We assume that the Trivial Mechanism has access to a test set of size $n$. Let $\alpha \in [0, 1]$,*

$$\varepsilon = \sqrt{\frac{4(k^{1-\alpha} \log(n + 1) + \log(2/\delta))}{n}},$$

*and $\eta = 1 - \frac{\varepsilon}{4(e^{\varepsilon k^\alpha} - 1)}$. If $N_\eta(\mathcal{F}) \leq (n + 1)^{k^{1-\alpha}}$, we have with probability $1 - \delta$*

$$\max_{1 \leq i \leq k} |\mathbb{E}_S[q_i] - \mathbb{E}_{\mathcal{D}}[q_i]| \leq \varepsilon, \tag{8}$$

*for any queries $q_1, q_2, \ldots q_k$ chosen adaptively by $\mathcal{A}$.*

*Proof.* Note that when $\eta = 1 - \frac{\varepsilon}{4(e^{\varepsilon k^\alpha} - 1)}$ we have $\log\left(1 + \frac{\varepsilon}{4(1-\eta)}\right) \geq \varepsilon k^\alpha$. Then, the result follows from the first part of Theorem 2. $\qquad \square$

In Corollary 5, the parameter $\alpha$ quantifies the strength of the similarity assumption. For $\alpha = 0$, there is no similarity requirement, and Corollary 5 always applies. In this case, the bound matches standard results for the trivial mechanism with $\varepsilon = \tilde{O}(\sqrt{k/n})$. However, as $\alpha$ grows, the similarity requirement becomes restrictive while the corresponding confidence interval becomes increasingly tight. In particular, for any $\alpha > 0$, if $\mathcal{F}$ permits a similarity cover $N_\eta(\mathcal{F}) \leq (n + 1)^{k^{1-\alpha}}$ for $\eta = 1 - (\varepsilon/4)(e^{\varepsilon k^\alpha} - 1)^{-1}$, we obtain a super linear improvement in the dependence on $k$. For instance, if $\alpha = 1/2$, then $\varepsilon = \tilde{O}(\sqrt{k^{1/2}/n})$, and we obtain a quadratic improvement in the number of queries for a fixed sample size. This improvement is similar to that achieved by the Gaussian mechanism [1, 5]. Moreover, since our technique is essentially tightening a union bound, this improvement easily extends to other mechanisms that rely on compression-based arguments, for instance, the Ladder Mechanism [2].

# 5 Empirical results

So far, we have established theoretically that similarity between classifiers allows us to evaluate a larger number of classifiers on the test set without overfitting. In this section, we investigate whether these improvements already occur in the regime of contemporary machine learning. We specifically focus on ImageNet and CIFAR-10, two widely used machine learning benchmarks that have recently been shown to exhibit little to no adaptive overfitting in spite of almost a decade of test set re-use [15]. For both datasets, we empirically measure two main quantities: (i) The similarity between a wide range of models, some of them arising from hyperparameter search experiments. (ii) The resulting increase in the number of models we can evaluate in a non-adaptive setting compared to a baseline that does not utilize the model similarities.

## 5.1 Similarities on Imagenet

We utilize the model testbed from Recht et al. [15],[1] who collected a dataset of 66 image classifiers that includes a wide range of standard ImageNet models such as AlexNet [10], ResNets [7], DenseNets [8], VGG [17], Inception [19], and several other models. As a baseline for the observed similarities between these models, we compare them to classifiers with the same accuracy but otherwise random predictions: given two models $f_1$ and $f_2$ with population error rates $\mu_1$ and $\mu_2$, we know that the similarity $\mathbb{P}(\mathbb{1}\{f_1(x) \neq y\} = \mathbb{1}\{f_2(x) \neq y\})$ equals $\mu_1\mu_2 + (1 - \mu_1)(1 - \mu_2)$ if the random variables $\mathbb{1}\{f_1(x) \neq y\}$ and $\mathbb{1}\{f_2(x) \neq y\}$ are independent. Figure 1a in the introduction shows these model similarities assuming the models make independent mistakes and also the empirical data for the $\binom{66}{2} = 2{,}145$ pairs of models. We see that the empirical similarities are significantly higher than the random baseline (mean 0.85 vs 0.62).

The corresponding Figure 1b shows two lower bounds on the number of models that can be evaluated for the empirical ImageNet data. In particular, we use $n = 50{,}000$ (the size of the ImageNet validation set) and a target probability $\delta = 0.05$ for the overfitting event (1) with error $\varepsilon = 0.01$. We compare two methods for computing the number of non-adaptively testable models: a guarantee based on the simple union bound (2) and a guarantee based on our more refined union bound derived from our theoretical analysis in Section 3. Later in this section, we introduce an even stronger bound that utilizes higher-order interactions between the model similarities and yields significantly larger improvements under an assumption on the structure among the classifiers.

To obtain meaningful quantities in the regime of ImageNet, all bounds here require significantly sharper numerical calculations than the standard theoretical tools such as Chernoff bounds. We now describe these calculations at a high level and defer the details to Appendix B. After introducing the three methods, we compare them on the ImageNet data.

**Standard union bound.** Given $n$, $\varepsilon$, and the population error rate of all models $\mathbb{E}_{\mathcal{D}}[q_i]$, we can compute the right hand side of (2) exactly.[2] It is well known that higher accuracies lead to smaller probability of error and hence allow for a larger number of test set reuses. We assume all models have population accuracy 75.6%, the average top-1 accuracy of the 66 Imagenet models. In this case, the vanilla union bound (2) guarantees that $k = 257{,}397$ models can be evaluated on a test set of size 50,000 so that their empirical accuracies would lie in the confidence interval $0.756 \pm 0.01$ with probability at least 95%.

**Similarity Union Bound.** While the union bound (2) is easy to use, it does not leverage the dependencies between the random variables $\mathbb{1}\{f_i(x) \neq y\}$ for $i \in \{1, 2, \ldots k\}$. To exploit this property, we utilize the refined union bound (3) which is guaranteed to be an improvement over (2) when the parameter $t$ is optimized. In order to use (3), we must compute the probabilities

$$\mathbb{P}\left(\{\mathbb{E}_S[q_2] - \mathbb{E}_{\mathcal{D}}[q_2] \leq \alpha_2\} \cap \{\mathbb{E}_S[q_1] - \mathbb{E}_{\mathcal{D}}[q_1] \geq \alpha_1\}\right) \tag{9}$$

for given $\alpha_1$, $\alpha_2$, $\mathbb{E}_{\mathcal{D}}[q_1]$, $\mathbb{E}_{\mathcal{D}}[q_2]$, and similarity $\mathbb{P}(q_1(z) = q_2(z))$. In Appendix B, we show that we can compute these probabilities efficiently by assigning success probabilities to three independent Bernoulli random variables $X_1$, $X_2$, and $W$ such that $(X_1 W, X_2 W)$ is equal to $(q_1(z), q_2(z))$ in

distribution. Let $p_w := \mathbb{P}(W = 1)$. Then, given i.i.d. draws $X_{1i}$, $X_{2i}$, and $W_i$, we condition on the values of $W_i$ to express probability (9) as

$$\mathbb{P}\left(\{\mathbb{E}_S[q_2] - \mathbb{E}_\mathcal{D}[q_2] \leq \alpha_2\} \cap \{\mathbb{E}_S[q_1] - \mathbb{E}_\mathcal{D}[q_1] \geq \alpha_1\}\right) \tag{10}$$

$$= \sum_{j=0}^{n} \binom{n}{j} p_w^j (1 - p_w)^{n-j} \mathbb{P}\left(\sum_{i=1}^{j} X_{2i} \leq \lfloor n(p_2 + \alpha_2) \rfloor\right) \mathbb{P}\left(\sum_{i=1}^{j} X_{1i} \geq \lceil n(p_1 + \alpha_1) \rceil\right).$$

We refer the reader to Appendix B for more details. The two tail probabilities for $X_{1i}$ and $X_{2i}$ can be computed efficiently with the use of beta functions. Using (10) and (3) with a binary search over $t$, we can compute the probability of making an error $\varepsilon$ when estimating the population error rates of $k$ models with given error rates and pairwise similarities. Figure 1b shows the maximum number of models $k$ that can be evaluated on the same test set so that the probability of making an $\varepsilon = 0.01$ error in estimating all their error rates is at most $0.05$ when the models satisfy $\mathbb{E}_\mathcal{D}[q_i] = 0.244$ and $\mathbb{P}(q_i(z) = q_j(z)) \geq 0.85$ for all $1 \leq i, j \leq k$. The figure shows that our new bound offers a significant improvement over the guarantee given by the standard union bound (2).

**Similarity union bound with a Naive Bayes assumption.** While the previous computation uses the pairwise similarities observed empirically to offer an improved guarantee on the number of allowed test set reuses, it does not take into account higher order dependencies between the models. In particular, Figure 4 in Appendix C shows that $27.8\%$ of test images are correctly classified by all the models, $55.9\%$ of test images are correctly classified by 60 of the 66 models considered, and $4.7\%$ of test images are incorrectly classified by all the models. We now show how this kind of agreement between models enables a larger number of test set reuses. Inspired by the coupling used in (10), we make the following assumption.

**Assumption A1** (Naive Bayes). *Let $q_1$, $q_2$, $\ldots q_k$ be a collection of queries such that $\mathbb{E}_\mathcal{D}[q_i] = p$ and $\mathbb{P}(q_i(z) = q_j(z)) = \eta$ for some $p$ and $\eta$, for all $1 \leq i, j \leq k$. We say such a collection has a Naive Bayes structure if there exist $p_x$ and $p_w$ in $[0, 1]$ such that $(q_1(z), q_2(z), \ldots, q_k(z))$ is equal to $(X_1 W, X_2 W, \ldots, X_k W)$ in distribution, where $W$, $X_1, \ldots X_k$ are independent Bernoulli random variables with $\mathbb{P}(W = 1) = p_w$ and $\mathbb{P}(X_i = 1) = p_x$ for all $1 \leq i \leq k$.*

Intuitively, a collection of queries $\mathbb{1}\{f_i(x) \neq y\}$ has a Naive Bayes structure if the data distribution $\mathcal{D}$ generates easy examples $(x, y)$ with probability $1 - p_w$ such that all the models $f_i$ classify correctly, and if an example is not easy, the models make mistakes independently. As mentioned before, Figure 4 supports the existence of such an easy set. When a test point in the ImageNet test set is not an easy example, the models do not make mistakes independently. Therefore, Assumption A1 is not exactly satisfied by existing ImageNet models. However, we know that independent Bernoulli trials saturate the standard union bound (2). This effect can also be observed in Figure 2. As the similarity between the models decreases, i.e. $1 - p_w$ decreases, the models make mistakes independently and the guarantee with Assumption A1 converges to the standard union bound guarantee. So while Assumption A1 is not exactly satisfied in practice, the violation among the ImageNet classifiers likely implies an even better lower bound on the number of testable models.

Assumption A1 is computationally advantageous. It allows us to compute the overfitting probability (1) exactly, as we detail in Appendix B. Figure 2 is an extension of Figure 1b; it shows the relative improvement of our bounds over the standard union bound in terms of the number of testable models when $\varepsilon = 0.01$ and $\delta = 0.01$. Moreover, Figure 2 also shows that the relative improvement of our bounds increases quickly with $\varepsilon$. According to Figure 2, Assumption A1 implies that we can evaluate $10^8$ models on the test set in the regime of ImageNet without overfitting. While this number of models might seem unnecessarily large, in Section 4 we saw that when models are chosen adaptively we must consider a tree of possible models, which can easily contain $10^8$ models.

## 5.2 Similarities on CIFAR-10

Practitioners often evaluate many more models than the handful that ultimately appear in publication. The choice of architecture is the result of a long period of iterative refinement, and the hyperparameters for any fixed architecture are often chosen by evaluating a large grid of plausible models. Using data from CIFAR-10, we demonstrate these common practices both generate large classes of very similar models.

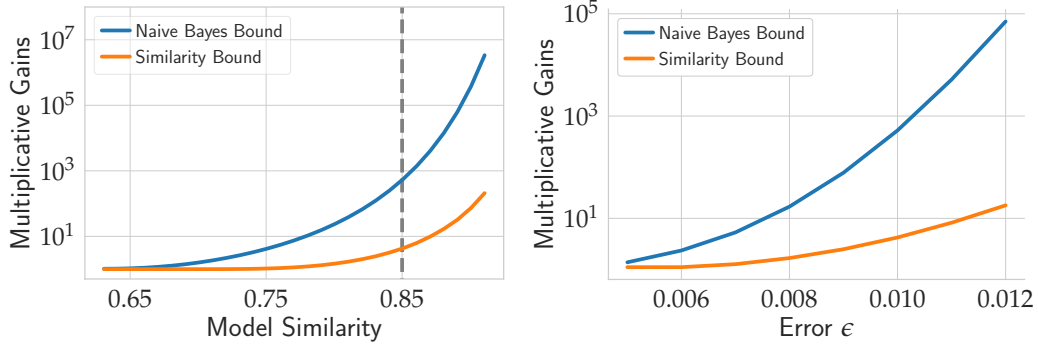

Figure 2: Left figure shows the multiplicative gains in the number of testable models, as a function of model similarity, over the guarantee offered by the standard union plus binomial bound, with $\varepsilon = 0.01$ and $\delta = 0.05$. Right figure shows the same multiplicative gains, but as a function of $\varepsilon$, when $\delta = 0.05$ and the pairwise similarity is $\eta = 0.85$.

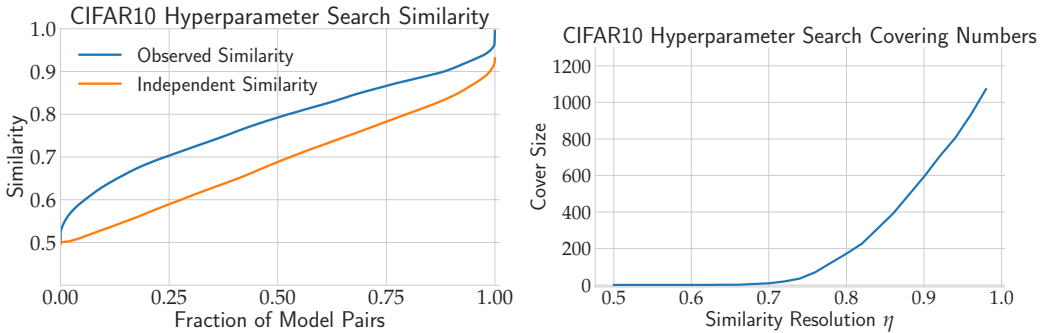

Figure 3: Model similarities and covering numbers for random hyperparameter search on CIFAR10.

**Random hyperparameter search.** To understand the similarity between models evaluated in hyperparameter search, we ran our own random search to choose hyperparameters for a ResNet-110. The grid included properties of the architecture (e.g. type of residual block), the optimization algorithm (e.g. choice of optimizer), and the data distribution (e.g. data augmentation strategies). A full specification of the grid is included in Appendix D. We sample and train 320 models, and, for each model, we select 10 checkpoints evenly spaced throughout training. The best model considered achieves accuracy of 96.6%, and, after restricting to models with accuracy at least 50%, we are left with 1,235 model checkpoints. In Figure 3, we show the similarity for each pair of checkpoints and compute an upper bound on the corresponding similarity covering number $N_\eta(\mathcal{F})$ for each possible value of $\eta$. As in the case of ImageNet, CIFAR10 models found by random search are significantly more similar than random chance would suggest.

**Neural architecture search.** In the random search experiment, all of the models were chosen non-adaptively—the grid of models is fixed in advance. However, similarity protects against overfitting also in the adaptive setting. To illustrate this, we compute the similarity for models evaluated by automatic neural architecture search. In particular, we ran the DARTS neural architecture search pipeline to adaptively evaluate a large number of plausible models in search of promising configurations [13, 14]. In Table 1, we report the mean accuracies and pairwise similarities for 20 randomly selected configurations evaluated by DARTS, as well as the top 20 scoring configurations according to DARTS internal scoring mechanism. Table 1 also shows the multiplicative gains in the number of testable models offered by our similarity bound (SB) and our naive Bayes bound (NBB) over the standard union bound are between one and four orders of magnitude. Therefore, even in a high accuracy regime we can guarantee a significantly higher number of test set reuses without overfitting when taking into account model similarities.

Table 1: Neural Architecture Search Similarities

| Models | Mean Accuracy | Mean Similarity | Increase in Testable Models | |
|---|---|---|---|---|
| | | | SB | NBB |
| 20 Random | 96.8% | 97.5% | **9.9**$\times$ | $\mathbf{1.6 \cdot 10^4}\times$ |
| 20 Highest Scoring | 96.9% | 97.6% | **12.0**$\times$ | $\mathbf{3.4 \cdot 10^4}\times$ |

## 6  Conclusions and future work

We have shown that contemporary image classification models are highly similar, and that this similarity increases the longevity of the test set both in theory and in experiment. It is worth noting that model similarity does not preclude progress on the test set: two models that are 85% similar can differ by as much as 15% in accuracy (for context: the top-5 accuracy improvement from the seminal AlexNet to the current state of the art on ImageNet is about 17%). In addition, it is well known that higher model accuracy implies a larger number of test set reuses without overfitting. So as the machine learning practitioner explores increasingly better performing models that also become more similar, it can actually become *harder* to overfit.

There are multiple important avenues for future work. First, one natural question is *why* the classification models turn out to be so similar. In addition, it would be insightful to understand whether the similarity phenomenon is specific to image classification or also arises in other classification tasks. There may also be further structural dependencies between models that mitigate the amount of overfitting. Finally, it would be ideal to have a statistical procedure that leverages such model structure to provide reliable and accurate performance bounds for test set re-use.

**Acknowledgements.**   We thank Vitaly Feldman for helpful discussions. This work is generously supported in part by ONR awards N00014-17-1-2191, N00014-17-1-2401, and N00014-18-1-2833, the DARPA Assured Autonomy (FA8750-18-C-0101) and Lagrange (W911NF-16-1-0552) programs, a Siemens Futuremakers Fellowship, an Amazon AWS AI Research Award, a gift from Microsoft Research, and the National Science Foundation Graduate Research Fellowship Program under Grant No. DGE 1752814.

## Footnotes

[1]Available at `https://github.com/modestyachts/ImageNetV2`.

[2]After an additional union bound to decouple the left and right tails.

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
