[Supplementary Material]

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

*Proof.* We assume $p_1 > 0$. The result follows by continuity when $p_1 = 0$. We prove the more general case since the first part of the lemma is a particular case. By standard Chernoff methods we have

$$\mathbb{P}\left(\frac{1}{n}\sum_{i=1}^{n} X_i > p_1 - p_{-1} + t\right) \leq e^{-n\lambda(t+p_1-p_{-1})}\left(p_0 + p_1 e^{\lambda} + p_{-1}e^{-\lambda}\right)^n,$$

for any $\lambda \in [0,\infty)$. Let $r > 0$ to be chosen later. Now, we would like to choose $\lambda$ to be nonnegative and as large as possible so that

$$p_0 + p_1 e^{\lambda} + p_{-1}e^{-\lambda} \leq e^{\lambda r}. \tag{11}$$

By changing variables to $e^{\lambda} = z + 1$ for some $z \geq 0$ we want to find $z$ as large as possible so that

$$p_0(z+1) + p_1(z+1)^2 + p_{-1} \leq (z+1)^{1+r}.$$

Then, by Bernoulli's inequality it suffices if $z$ satisfies the inequality

$$p_0(z+1) + p_1(z+1)^2 + p_{-1} \leq 1 + (1+r)\,z,$$

which is equivalent to

$$p_0 + p_1 z + 2p_1 \leq 1 + r.$$

Hence, the desired inequality (11) is satisfied if $z \leq \frac{p_{-1}-p_1+r}{p_1}$, which can be satisfied by choosing $z = \frac{p_{-1}-p_1+r}{p_1}$ when $p_{-1} - p_1 + r \geq 0$. In this case, we would be able to set $\lambda = \log\left(1 + \frac{p_{-1}-p_1+r}{p_1}\right)$ and obtain

$$\mathbb{P}\left(\frac{1}{n}\sum_{i=1}^{n} X_i > p_1 - p_{-1} + t\right) \leq e^{-n\log\left(1+\frac{p_{-1}-p_1+r}{p_1}\right)(t+p_1-p_{-1}-r)}.$$

Set $r = p_1 - p_{-1} + t/2$ and by the assumption on $t$ we are guaranteed that $r \geq 0$ and $p_{-1}-p_1+r \geq 0$. The conclusion follows. $\qquad\square$

**Theorem 2.** *Let $\mathcal{F} = \{q_1, q_2, \ldots, q_k\}$ be a collection of queries $q_i\colon \mathcal{Z} \to \{0,1\}$ independent of the test set $\{z_1, z_2, \ldots, z_n\}$. Then, for any $\eta \in [0,1]$ we have*

$$\mathbb{P}\left(\max_{1\leq i\leq k}|\mathbb{E}_S[q_i] - \mathbb{E}_{\mathcal{D}}[q_i]| \geq \varepsilon\right) \leq 2N_\eta e^{-\frac{n\varepsilon^2}{2}} + 2ke^{-\frac{n\varepsilon}{4}\log\left(1+\frac{\varepsilon}{4(1-\eta)}\right)}. \tag{4}$$

*Then, for all $\eta \leq 1 - \max\left\{\frac{2\log(4k/\delta)}{n}, \sqrt{\frac{\log(4N_\eta/\delta)}{2n}}\right\}$, we have with probability $1-\delta$*

$$\max_{1\leq i\leq k}|\mathbb{E}_S[q_i] - \mathbb{E}_{\mathcal{D}}[q_i]| \leq \max\left\{\sqrt{\frac{2\log(4N_\eta/\delta)}{n}}, \sqrt{\frac{32(1-\eta)\log(4k/\delta)}{n}}\right\}. \tag{5}$$

*Moreover, if $\varepsilon = \sqrt{\frac{\log((2N_\eta+1)/\delta)}{n}}$ and $\eta \geq 1 - \frac{\varepsilon}{4\left(e^{2\varepsilon}(2k)^{\frac{4}{n\varepsilon}}-1\right)}$, we have with probability $1-\delta$*

$$\max_{1\leq i\leq k}|\mathbb{E}_S[q_i] - \mathbb{E}_{\mathcal{D}}[q_i]| \leq \varepsilon. \tag{6}$$

*Proof.* First we prove (4) and we start with the right tails. We have

$$\mathbb{P}\left(\bigcup_{i=1}^{k}\{\mathbb{E}_S[q_i] - \mathbb{E}_{\mathcal{D}}[q_i] \geq \varepsilon\}\right) \leq \mathbb{P}\left(\bigcup_{i=1}^{k}\{\mathbb{E}_S[q_i] - \mathbb{E}_{\mathcal{D}}[q_i] \geq \varepsilon\} \bigcup_{\widetilde{q} \in M}\{\mathbb{E}_S[\widetilde{q}] - \mathbb{E}_{\mathcal{D}}[\widetilde{q}] \geq \varepsilon\}\right),$$

where $M$ is a minimal $\eta$ similarity cover of $\mathcal{F}$. Then, there exists a partition of $\mathcal{F}$ into subsets $R_{\widetilde{q}}$, with $\widetilde{q} \in M$, such that for any $q \in \mathcal{F}$ there exists $\widetilde{q}$ such that $q \in R_{\widetilde{q}}$, $\mathbb{E}_{\mathcal{D}}[q] \geq \mathbb{E}_{\mathcal{D}}[\widetilde{q}]$, and $\mathbb{P}(q(z) = \widetilde{q}(z)) \geq \eta$. Since $R_{\widetilde{q}}$ is a partition of $\mathcal{F}$, we have $\sum_{\widetilde{q} \in M} |R_{\widetilde{q}}| = k$. Therefore, following the same argument as in (3), we have

$$\mathbb{P}\left(\bigcup_{i=1}^{k}\{\mathbb{E}_S[q_i] - \mathbb{E}_{\mathcal{D}}[q_i] \geq \varepsilon\}\right) \leq \sum_{\widetilde{q} \in M} \mathbb{P}\left(\mathbb{E}_S[\widetilde{q}] - \mathbb{E}_{\mathcal{D}}[\widetilde{q}] \geq \frac{\varepsilon}{2}\right)$$

$$+ \sum_{\widetilde{q} \in M} \sum_{q \in R_{\widetilde{q}}} \mathbb{P}\left(\{\mathbb{E}_S[\widetilde{q}] - \mathbb{E}_{\mathcal{D}}[\widetilde{q}] \leq \varepsilon/2\} \cap \{\mathbb{E}_S[q] - \mathbb{E}_{\mathcal{D}}[q] \geq \varepsilon\}\right)$$

$$\leq \sum_{\widetilde{q} \in M} \mathbb{P}\left(\mathbb{E}_S[\widetilde{q}] - \mathbb{E}_{\mathcal{D}}[\widetilde{q}] \geq \frac{\varepsilon}{2}\right)$$

$$+ \sum_{\widetilde{q} \in M} \sum_{q \in R_{\widetilde{q}}} \mathbb{P}\left(\mathbb{E}_S[\widetilde{q}] - \mathbb{E}_{\mathcal{D}}[\widetilde{q}] + \varepsilon/2 \leq \mathbb{E}_S[q] - \mathbb{E}_{\mathcal{D}}[q]\right).$$

Now, for every $\widetilde{q} \in M$ and any $q \in R_{\widetilde{q}}$ we use a standard Chernoff bound and Lemma 4 to show

$$\mathbb{P}\left(\mathbb{E}_S[\widetilde{q}] - \mathbb{E}_{\mathcal{D}}[\widetilde{q}] \geq \frac{\varepsilon}{2}\right) \leq e^{-\frac{n\varepsilon^2}{2}} \text{ and } \mathbb{P}\left(\mathbb{E}_S[\widetilde{q}] - \mathbb{E}_{\mathcal{D}}[\widetilde{q}] + \varepsilon/2 \leq \mathbb{E}_S[q] - \mathbb{E}_{\mathcal{D}}[q]\right) \leq e^{-\frac{n\varepsilon}{4}\log\left(1 + \frac{\varepsilon}{4(1-\eta)}\right)}.$$

To see why we can apply Lemma 4 note that $q(z) - \widetilde{q}(z)$ takes values in $\{-1, 0, 1\}$ with the probability of 0 being at least $\eta$, and $\mathbb{E}_{\mathcal{D}}[q - \widetilde{q}] \geq 0$ by the choice of the covering set. Since $|M| = N_\eta$ and since $\sum_{\widetilde{q} \in M} |R_{\widetilde{q}}| = k$, we find

$$\mathbb{P}\left(\bigcup_{i=1}^{k}\{\mathbb{E}_S[q_i] - \mathbb{E}_{\mathcal{D}}[q_i] \geq \varepsilon\}\right) \leq N_\eta e^{-\frac{n\varepsilon^2}{2}} + k e^{-\frac{n\varepsilon}{4}\log\left(1 + \frac{\varepsilon}{4(1-\eta)}\right)}.$$

An analogous argument for the left tails yields (4). Now, we turn to showing (5). The goal is to find $\varepsilon$ such that

$$2N_\eta e^{-\frac{n\varepsilon^2}{2}} \leq \frac{\delta}{2} \text{ and } 2k e^{-\frac{n\varepsilon}{4}\log\left(1 + \frac{\varepsilon}{4(1-\eta)}\right)} \leq \frac{\delta}{2}. \tag{12}$$

The first inequality is satisfied if $\varepsilon \geq \sqrt{\frac{2\log(4N_\eta/\delta)}{n}}$. To find $\varepsilon$ that satisfies the second condition we make use of the inequality $\log(1 + t) \geq \frac{t}{t+1}$ for all $t \geq 0$. We search for $\varepsilon$ that also satisfies $\varepsilon \leq 4(1 - \eta)$. Then,

$$\frac{n\varepsilon}{4}\log\left(1 + \frac{\varepsilon}{4(1-\eta)}\right) \geq \frac{n\varepsilon^2}{32(1-\eta)},$$

and we would like the right hand side to be at least $\log(4k/\delta)$. If we choose

$$\varepsilon = \max\left\{\sqrt{\frac{2\log(4N_\eta/\delta)}{n}}, \sqrt{\frac{32(1-\eta)\log(4k/\delta)}{n}}\right\},$$

the condition $\varepsilon \leq 4(1 - \eta)$ is satisfied because of the assumption on $\eta$. In this case, both conditions (12) are satisfied and (5) is proven. Finally, note that when $\eta \geq 1 - \frac{\varepsilon}{4(e^{2\varepsilon}(2k)^{\frac{4}{n\varepsilon}} - 1)}$ we have

$$2k e^{-\frac{n\varepsilon}{4}\log\left(1 + \frac{\varepsilon}{4(1-\eta)}\right)} \leq e^{-\frac{n\varepsilon^2}{2}}.$$

Then, (6) follows by choosing $\varepsilon = \sqrt{\frac{\log((2N_\eta+1)/\delta)}{n}}$. This completes the proof.

$\square$

**Corollary 3.** *Let $\mathcal{F} = \{q_1, q_2, \ldots, q_k\}$ be a collection of $k$ queries $q_i \colon \mathcal{Z} \to \{0,1\}$ fixed independently of the test set. Choose $\eta_\star$ so that $N_{\eta_\star} = 1$. Suppose $n \geq c_1 \max\{\varepsilon^{-1}, \varepsilon^{-2}\}$ and the number of queries $k$ satisfies*

$$k \leq \frac{c_2 \varepsilon}{(1 - \eta_\star)} \tag{7}$$

*for positive constants $c_1, c_2$. Then, with probability $3/4$, $\max_{1 \leq i \leq k} |\mathbb{E}_S[q_i] - \mathbb{E}_{\mathcal{D}}[q_i]| \leq \varepsilon$.*

*Proof.* Choose $\eta_\star$ so that $N_{\eta_\star} = 1$. Using equation (4) from Theorem (2),

$$\mathbb{P}\left(\max_{1 \leq i \leq k} |\mathbb{E}_S[q_i] - \mathbb{E}_{\mathcal{D}}[q_i]| \geq \varepsilon\right) \leq 2N_\eta e^{-\frac{n\varepsilon^2}{2}} + 2ke^{-\frac{n\varepsilon}{4}\log\left(1 + \frac{\varepsilon}{4(1-\eta)}\right)}.$$

Consider each term separately. If $n \geq \frac{\log(4/\delta)}{\varepsilon^2}$, then the first term $2N_{\eta_\star}e^{-n\varepsilon^2/2} \leq \delta/2$. Now, we choose $k$ so that the second term is at most $\delta/2$, i.e.

$$2ke^{-\frac{n\varepsilon}{4}\log\left(1 + \frac{\varepsilon}{4(1-\eta_\star)}\right)} \leq \frac{\delta}{2},$$

which is equivalent to requiring

$$k \leq \frac{\delta}{4}\left(1 + \frac{\varepsilon}{4(1-\eta_\star)}\right)^{n\varepsilon/4}.$$

If $n \geq \frac{4}{\epsilon}$, then the right hand side can be lower bounded by

$$\frac{\delta}{4}\left(1 + \frac{\varepsilon}{4(1-\eta)}\right)^{n\varepsilon/4} \geq \frac{\delta}{4}\left(1 + \frac{\varepsilon}{4(1-\eta_\star)}\right) \geq \frac{\delta}{4}\left(\frac{\varepsilon}{4(1-\eta_\star)}\right),$$

and the conclusion follows from plugging in $\delta = 1/4$. $\qquad\square$

# B  Tail probability of two dependent binomials

In this section we detail the computations of the two similarity union bounds (with and without the Naive Bayes assumption).

**Similarity Union Bound.**  We wish to compute the probability

$$\mathbb{P}\left(\{\mathbb{E}_S[q_2] - \mathbb{E}_{\mathcal{D}}[q_2] \leq \alpha_2\} \cap \{\mathbb{E}_S[q_1] - \mathbb{E}_{\mathcal{D}}[q_1] \geq \alpha_1\}\right), \tag{13}$$

where $q_1(z)$ and $q_2(z)$ have some joint distribution over $\{0,1\}^2$. Let use denote $p_1 = \mathbb{E}_{\mathcal{D}}[q_1]$, $p_2 = \mathbb{E}_{\mathcal{D}}[q_2]$, and $\eta = \mathbb{P}(q_1(z) = q_2(z))$ respectively. These three quantities fully determine the joint probability distribution of $q_1(z)$ and $q_2(z)$. Specifically, we have

$$\mathbb{P}(q_1(z) = 1, q_q(z) = 1) = \frac{p_1 + p_2 + \eta - 1}{2}, \quad \mathbb{P}(q_1(z) = 1, q_q(z) = 0) = \frac{1 + p_1 - p_2 - \eta}{2}$$

$$\mathbb{P}(q_1(z) = 0, q_q(z) = 1) = \frac{1 + p_2 - p_1 - \eta}{2}, \quad \mathbb{P}(q_1(z) = 0, q_q(z) = 0) = \frac{1 + \eta - p_1 - p_2}{2}.$$

We denote these four probabilities by $p_{11}$, $p_{10}$, $p_{01}$, and $p_{00}$ respectively. We aim to find three independent Bernoulli random variables $X_1$, $X_2$, and $W$ such that $(X_1W, X_2W)$ equals $(q_1(z), q_2(z))$ in distribution. It turns out we can achieve this whenever $p_{11} \geq (p_{10} + p_{11})(p_{01} + p_{11})$, a condition that is always satisfied in the settings we consider, by setting

$$\mathbb{P}(X_1 = 1) = \frac{p_{11}}{p_{01} + p_{11}}, \quad \mathbb{P}(X_2 = 1) = \frac{p_{11}}{p_{10} + p_{11}}, \quad \mathbb{P}(W = 1) = \frac{(p_{10} + p_{11})(p_{01} + p_{11})}{p_{11}}.$$

Then, given i.i.d. draws $X_{1i}$, $X_{2i}$, and $W_i$, probability (9) equals

$$\mathbb{P}\left(\left\{\sum_{i=1}^n X_{2i}W_i \leq \lfloor n(p_2 + \alpha_2)\rfloor\right\} \cap \left\{\sum_{i=1}^n X_{1i}W_i \geq \lceil n(p_1 + \alpha_1)\rceil\right\}\right). \tag{14}$$

Denote $p_w = \mathbb{P}(W = 1)$. Then, we condition on the possible values of $W_i$ to obtain

$$\mathbb{P}\left(\{\mathbb{E}_S[q_2] - \mathbb{E}_{\mathcal{D}}[q_2] \leq \alpha_2\} \cap \{\mathbb{E}_S[q_1] - \mathbb{E}_{\mathcal{D}}[q_1] \geq \alpha_1\}\right) \tag{15}$$

$$= \sum_{j=0}^n \binom{n}{j} p_w^j (1 - p_w)^{n-j} \mathbb{P}\left(\sum_{i=1}^j X_{2i} \leq \lfloor n(p_2 + \alpha_2)\rfloor\right) \mathbb{P}\left(\sum_{i=1}^j X_{1i} \geq \lceil n(p_1 + \alpha_1)\rceil\right).$$

The two tail probabilities for $X_{1i}$ and $X_{2i}$ can be computed efficiently with the use of beta functions.

**Similarity union bound with a Naive Bayes assumption.** In this section we wish to compute directly the overfitting probability

$$\mathbb{P}\left(\max_{1 \leq i \leq k} |\mathbb{E}_S[q_i] - \mathbb{E}_\mathcal{D}[q_i]| \geq \varepsilon\right) \tag{16}$$

when the query vector $(q_1(z), q_2(z), \ldots, q_k(z))$ is equal in distribution to $(X_1 W, X_2 W, \ldots, X_k W)$ for some independent Bernoulli random variables $W, X_1, \ldots X_k$. Recall that we assume that all queries $q_i$ have equal error rates $\mathbb{E}_\mathcal{D}[q_i]$; let us denote it $\mu = \mathbb{E}_\mathcal{D}[q_i]$. Moreover, for any two queries $q_i$ and $q_j$ we have $\mathbb{P}(q_i(z) = q_j(z)) = \eta$.

Suppose we are given i.i.d. draws $W_i$ and i.i.d. draws $X_{\ell i}$ for $1 \leq i \leq n$ and $1 \leq \ell \leq k$. Then, if $p_w := \mathbb{P}(W = 1)$, by conditioning on the values of the random variables $W_i$ we obtain

$$\mathbb{P}\left(\max_{1 \leq i \leq k} |\mathbb{E}_S[q_i] - \mathbb{E}_\mathcal{D}[q_i]| \geq \varepsilon\right) = \sum_{j=1}^{n} \binom{n}{j} p_w^j (1-p_w)^{n-j} \mathbb{P}\left(\bigcup_{\ell=1}^{k} \left|\frac{1}{n} \sum_{i=1}^{j} X_{\ell i} - \mu\right| \geq \varepsilon\right).$$

The random variables $\sum_{i=1}^{j} X_{\ell i}$ have the same distribution for all $\ell$ and are independent. Then,

$$\mathbb{P}\left(\bigcup_{\ell=1}^{k} \left|\frac{1}{n} \sum_{i=1}^{j} X_{\ell i} - \mu\right| \geq \varepsilon\right) = 1 - \mathbb{P}\left(\bigcap_{\ell=1}^{k} \left|\frac{1}{n} \sum_{i=1}^{j} X_{\ell i} - \mu\right| < \varepsilon\right)$$

$$= 1 - \mathbb{P}\left(\left|\frac{1}{n} \sum_{i=1}^{j} X_{1i} - \mu\right| < \varepsilon\right)^k.$$

Therefore, we have

$$\mathbb{P}\left(\max_{1 \leq i \leq k} |\mathbb{E}_S[q_i] - \mathbb{E}_\mathcal{D}[q_i]| \geq \varepsilon\right) = \sum_{j=1}^{n} \binom{n}{j} p_w^j (1-p_w)^{n-j} \left[1 - \mathbb{P}\left(\left|\frac{1}{n} \sum_{i=1}^{j} X_{1i} - \mu\right| < \varepsilon\right)^k\right].$$

## C  Empirical distribution of image difficulty in ImageNet

Figure 4: The empirical "difficulty" distribution of the 50,000 images in the ImageNet validation set as measured by the classifiers in the testbed from Recht et al. [15]. The plot shows how many of the images are misclassified by at most a certain number of the models. For instance, about 27.8% of the images are correctly classified by all models, and 55.9% of the images are correctly classified by 60 of the 66 models. 4.7% of the images are misclassified by all models. The plot shows that a significant fraction of images is classified correctly by all or almost all of the models. These empirical findings support the Naive Bayes assumption in Section 5.1.

Table 2: Random grid search hyperparameters.

| Parameter | Sampling Distribution |
|---|---|
| Number of base channels | Uniform$\{4, 8, 16, 32\}$ |
| Residual block type | Uniform$\{$"Basic", "Bottleneck"$\}$ |
| Remove ReLu before residual units | Uniform$\{$True, False$\}$ |
| Add BatchNorm after last convolutions | Uniform$\{$True, False$\}$ |
| Preactivation of shortcuts after downsampling | Uniform$\{$True, False$\}$ |
| Batch size | Uniform$\{32, 64, 128, 256\}$ |
| Base learning rate | Uniform[1e-4, 0.5] |
| Weight decay | $10^{\text{Uniform}[-5,-1]}$ |
| Use weight decay with batch norm | Uniform$\{$True, False$\}$ |
| Optimizer | Uniform$\{$SGD, SGD with Momentum, Nesterov GD, Adam$\}$ |
| Momentum (SGD with momentum) | Uniform$\{0.6, 0.99\}$ |
| $\beta_1$ (Adam) | Uniform[0.8, 0.95] |
| $\beta_2$ (Adam) | Uniform[0.9, 0.999] |
| Learning rate schedule | Uniform$\{$Cosine, Fixed Decay$\}$ |
| Learning rate decay point 1 (Fixed Decay) | Uniform$\{40, 60, 80, 100\}$ |
| Learning rate decay point 2 (Fixed Decay) | Uniform$\{120, 140, 160, 180\}$ |
| Use random crops | Uniform$\{$True, False$\}$ |
| Random crop padding | Uniform$\{2, 4, 8\}$ |
| Use horizontal flips | Uniform$\{$True, False$\}$ |
| Use cutout | Uniform$\{$True, False$\}$ |
| Cutout size | Uniform$\{8, 12, 16\}$ |
| Use dual cutout augmentation | Uniform$\{$True, False$\}$ |
| Dual cutout $\alpha$ | Uniform[0.05, 0.3] |
| Use random erasing | Uniform$\{$True, False$\}$ |
| Random erasing probability | Uniform[0.2, 0.8] |
| Use mixup data augmentation | Uniform$\{$True, False$\}$ |
| Mixup $\alpha$ | Uniform[0.6, 1.4] |
| Use label smoothing | Uniform$\{$True, False$\}$ |
| Label smoothing $\epsilon$ | Uniform[0.01, 0.2] |

# D    CIFAR-10 random hyperparameter grid search

We conducted a large hyperparameter search on a ResNet110. All of our experiments build on the ResNet implementation and training code provided by `https://github.com/hysts/pytorch_image_classification`. In Table 2, we specify the grid used in the experiments. If not explicitly stated, all other hyperparameters are set to their default settings.