[Reviews · NeurIPS 2019]

Reviewer 1



With my limited familiarity with the area, I found the paper to be easy and enjoyable to read and follow. The authors did a good job of outlining the motivation, existing work, and limitations. The work seems like a strong combination of an interesting theoretical finding, which requires a neat extension of the union bound as explained in Section 3, and an assessment of its empirical impact. The problem addressed -- namely, quantifying the effect of test set overuse -- is very timely, and the result that model similarity provably increases the number of times one can re-use a test set is quite interesting. Given the complex expressions in Theorem 2 and Corollary 4, it would be helpful to either also provide a simplified form (under further assumptions) or analyze the results for some interesting special cases for which the expressions become simpler and dependencies on parameters more apparent. For the adaptive classification case (Section 4), the analysis using (n+1)^{k-1} as the maximum number of queries seems quite conservative. Could you comment on its tightness / usefulness in practice? For corollary 4, it seems the way the parameters (alpha, eps, etc.) are stated may not be the most natural or useful. It is not clear when the corollary provides a useful result. E.g., as stated, eps is a function of alpha, eta is a function of eps and alpha, and N_eta is required to be bounded by a function of alpha. This makes it difficult to assess simple things like whether alpha has a positive or negative effect on the result. Would it help to re-state the result in terms of (only) the largest alpha that satisfies the bound on N_eta? The Naive Bayes assumption that comes towards the end of page 6 is left mysterious till that point in the paper. I would suggest alluding to it (or at least its textual interpretation near line 237) early on in the paper.

Reviewer 2



The writing focuses on mathematical analysis of the case where all model predictions tend to be correlated as well as on empirical analysis of how correlated the model predictions are in practice. The math looks OK, but,it is the straightforward empirical analysis that is most valuable, and it seems to me that all of the theoretical conclusions could be made without theory, simply through simulation instead of work on deriving bounds. Thus, it is hard for me to see value in large parts of the paper.

Reviewer 3



It has been previously suggested that the currently massive reuse of common test data sets can lead to overfitting of machine learning models. This work provides new analysis and support for this observation that the similarity between alternative models, following from shared prior knowledge and modeling choices, can reduce the potential overfitting substantially when compared to the theoretical expectations. The paper demonstrates empirically that predictions provided by alternative models are more similar than what could be expected just based on the prediction accuracy and provide an exact quantification of this effect. The authors use this observation to derive modified confidence interval for predictions that incorporates the similarity information. Application is demonstrated with real data. Quality. The work provides new information on is of good quality; the derivations follow standard style. A weak point is that the discussion is limited to relatively specific data sets / modeling tasks. Earlier work is being appropriately discussed. Clarity. Overall the text is well written and clear to follow. A shortcoming is that the Introduction does not include any references, and existing theoretical results are mentioned without citation (e.g. rows 19-21). Originality. The idea that overfitting is mitigated thourh similarity between models is not novel as such but compared to earlier related approaches that are being cited, this work provides new empirical evidence and theoretical understanding of this phenomenon. Significance. The results are useful for understanding fundamental aspects and limitations of machine learning techniques, and the pragmatically motivated, improved error bounds can be useful in practical applications. The vast applicability across a range of sample sizes, data set types, and modeling tasks remains to be shown, however.

[Author Response · NeurIPS 2019]

We thank the reviewers for their positive feedback and comments. We will incorporate their suggestions into the next version of the paper. We address points made by each individual reviewer in turn.

**Reviewer 1**

- In the revision, we will present simplifications of Theorem 2 and Corollary 4. For instance, if we fix $n, \delta$, and $\epsilon$ and choose $\eta$ so that $N_\eta = 1$ (i.e., at least one model is $\eta$-similar to all others), then Theorem 2 bounds the number of testable models as
$$k \leq \frac{c_1}{(1-\eta)^{c_2}},$$
where $c_1, c_2 \geq 0$ are constants that depend on $n, \delta$, and $\epsilon$. Then, as the similarity $\eta$ of the model collection grows, the number of testable models $k$ grows as well.

- In Corollary 4, $\alpha$ quantifies the strength of the similarity assumption. For $\alpha = 0$, there is no similarity requirement, and the confidence interval is wide. However, as $\alpha$ grows, the similarity requirement becomes restrictive while the confidence interval becomes increasingly tight. We will make this dependence clear and simplify the stated bound.

- In the adaptive query case, the $(n+1)^{k-1}$ bound on the number of queries is standard in the adaptive data analysis literature. The bound is indeed conservative and reflects the worst-case behavior of the analyst. Our analysis helps illustrate why this worst-case behavior does not attain in practice. If the queries are similar, then the effective number of queries can be much smaller than $(n+1)^{k-1}$.

**Reviewer 2**

We begin with the main point of Reviewer 2. The reviewer is concerned that the paper's conclusions could be derived solely from simulation work rather than mathematical analysis. Simply put, **the $y$-axes of our plots heavily rely upon our theoretical contributions.** As a result, we are not aware of a way to derive our paper's conclusions purely from simulation. Since the review is not specific about what types of simulations would yield similar insights into overfitting, we would appreciate a clarification in the updated review.

To expand on this further: the bulk of the theoretical content is careful numerical calculations to obtain sharp generalization bounds where the improvement due to similar models is apparent on data from ImageNet and CIFAR-10. The standard union bound does not take advantage of model similarity; the refinement in equation (3) is the basis for our calculations that highlight the beneficial effect of model similarity. These sharp bounds allow us to empirically demonstrate that model similarity offers protection against overfitting on heavily used benchmark datasets. Consequently, the mathematical analysis is not merely an accompaniment to our empirical results, but an important part in their development.

Furthermore, generalization bounds of the type we present in Theorem 2 and Corollary 4 are a core focus in statistical learning theory. Our bounds demonstrate a link between similarity and protection against overfitting in both the non-adaptive and adaptive case. The settings considered are purposefully simple to highlight the main ideas, and, as suggested by Reviewer 1, we will further simplify the bounds to improve their pedagogic value. Finally, generalization bounds can often be vacuous when instantiated with concrete values from practical settings. A key contribution of our work is generalization bounds that provide meaningful numerical guarantees on both ImageNet and CIFAR-10 data.

We now address the remaining comments of Reviewer 2:

- We agree an important future direction is to expand the scope of benchmarks considered. We plan to conduct the same similarity analysis on both Kaggle competitions and tasks in natural language processing.

- Our analysis explicitly considers the case where models cluster into smaller sets with large $\eta$. The definition of an *$\eta$-similarity cover* is a clustering such that each cluster has similarity at least $\eta$. Figure 3 shows one such example: for $\eta \approx 0.8$, the models can be partitioned into roughly 200 clusters.

- Thank you for pointing out the typo for $p_w$. In lines 238 and 244, $p_w$ should be replaced with $1 - p_w$. Otherwise, the definitions and subsequent discussion are unchanged.

**Reviewer 3**

- We agree an important direction for future work is exploring the extent to which our findings transfer to other settings. A concrete next step is evaluating model similarity on data from Kaggle competitions, which includes a diverse set of sample sizes, model classes, and data types. Nevertheless, image classification on ImageNet and CIFAR-10 is a natural starting place for this line of inquiry. Recent research has shown that both benchmarks do not suffer from adaptive overfitting despite almost a decade of intense activity, and model similarity offers an important piece of the explanation for this surprising phenomenon.

- We thank the reviewer for pointing out the lack of citations in the introduction. We will update the introduction with appropriate citations. We would appreciate any pointers to additional related work in the updated review.

[Meta-Review · NeurIPS 2019]

This paper is concerned with an observation about adaptive data analysis. It relies on a study that shows that despite statistical lower bounds, common practices of adaptive data analysis do not result in overfitting. The authors show that empirically this is a result of the models used in Kaggle competitions behaving in a similar manner. In addition, the authors give a simple model and analyze the model. The reviewers thought this is an interesting direction and that the results were generally well executed. This paper was thoroughly discussed by the committee, and was discussed with the PC Chair. A topic that came into discussion by the committee was the tight relationship between this paper and paper: 4929 A Meta-Analysis of Overfitting in Machine Learning which provides a large scale analysis of overfitting, similar to that performed by Recht et al. ICML 2019. The committee thought that paper 5286 provides the methodological progress expected for a contribution to NeurIPS, given the contribution by Recht et al. ICML 2019. After a great deal of deliberation, the committee thought that there is no room to have both paper accepted, and between paper 4929 and 5286, paper 5286 should be accepted. Recommendation: Throughout the process, multiple committee members read both papers 4929 and paper 5286. Some of the senior committee members were aware of the fact there is significant overlap in the papers’ author list. It seems that work in 5286 is strengthened by findings in 4929. All members of the committee thought that merging the two papers would result in a much stronger result and statement. The committee decided it will not force this solution, but recommend it to the authors.